

# Transcultural adaptation and validation of the questionnaire "Urgency, Weak stream, Incomplete emptying and Nocturia (UWIN)" for the Brazilian Portuguese

Caroline S. Silva[1], Katia S. Freitas[1], Anna Paloma R. Ribeiro[1], Cristiano M. Gomes[2] and Jose Bessa Junior[1,3]

[1] Department of Public Health, State University of Feira de Santana, Feira de Santana, Bahia, Brazil
[2] Division of Urology, Medical School, University of São Paulo, Sao Paulo, Sao Paulo, Brazil
[3] Division of Urology, Medical School, State University of Feira de Santana, Feira de Santana, Bahia, Brazil

Corresponding author
Jose Bessa Junior, bessa@uefs.br

## ABSTRACT

Lower urinary tract symptoms (LUTS) are common complaints in the adult male population and have a negative affect in the quality of life and represent an economic burden for the healthcare system worldwide. The International Prostatism Symptom Score (IPSS) is a validated tool for assessing these symptoms, but patients with low literacy may have difficulties comprehending and completing it accurately. The Urgency, Weak Stream, Incomplete Emptying, and Nocturia (UWIN) questionnaire was developed as a shorter tool in the assessment of LUTS to improve accuracy and minimize error. This study aimed at performing cross-cultural adaptation to Brazilian Portuguese and validation of UWIN questionnaire for patients with LUTS. The cross-cultural adaptation followed the steps of conceptual, item, semantic, operational, and pre-test equivalence to obtain the UWIN-Br version. The IPSS (gold standard) and UWIN-Br were coadministered, and information regarding the uroflowmetry examination was also recorded. We evaluated 306 men, median age 59 [52-66] years. There was a positive correlation $r = 0.804$ ($p < 0.001$) between the total IPSS score and the total UWIN-Br score, as well as the quality of life question ($r = 0.761$) ($p < 0.001$). The Bland-Altman plot showed good agreement between. Also, we observed that the maximum flow values decreased with the severity of the LUTS. UWIN-Br demonstrated excellent diagnostic accuracy in detecting the most severe cases. The area under the ROC curve was 89% [81–97%] 95% CI. 280 (91.5%) subjects completed the questionnaires without help, while 26 (8.5%) did so through an interview. The response time was 1.2 [1.0–1.5] min for UWIN-Br and 2.8 [2.2–3.4] min for IPSS ($p < 0.001$). UWIN-Br presents satisfactory and similar psychometric properties to the IPSS in the evaluation of LUTS and quality of life and is suitable for use in both clinical practice and research in our country.

## INTRODUCTION

Lower urinary tract symptoms (LUTS) is a broad term that describes storage, voiding, and postmicturition symptoms, according to the definition of the International Continence Society. The prevalence of LUTS in men older than 40 years of age exceeds 60% and increases with age (*Irwin et al., 2006*; *Coyne et al., 2009*; *Soler et al., 2018*). LUTS may be very bothersome to patients and may have a negative impact on work productivity, health-related quality of life, and social functioning (*Coyne et al., 2008*; *Coyne et al., 2011*).

The initial evaluation of men with LUTS includes history and physical examination, urinalysis, a subjective assessment with a survey of symptoms and objective tests, including uroflowmetry and measurement of postvoid residual urine volume (*Gravas et al., 2015*).

The International Prostate Symptom Score (IPSS) is a validated, well-established instrument, which enables the stratification of patients according to the severity of symptoms and allows for the assessment of both responses to treatment and disease progression. In Brazil, the instrument was validated by Berger and Cols in 1999, from the application of the same group of 281 participants with benign prostatic hyperplasia. However, patients with a low education level have been demonstrated to report difficulty completing it accurately (*Johnson et al., 2008*).

Simplified questionnaires have been developed and validated and may be more appropriate for the evaluation of subjects with lower education level (*Van Der Walt et al., 2011*; *Eid et al., 2014*; *Stothers et al., 2017*; *Mallya et al., 2017*; *Moses et al., 2017*). Among these instruments is the UWIN (Urgency, Weak Stream, Incomplete Emptying and Nocturia), which was developed to facilitate the completion of the IPSS questionnaire, thus making the administration of the questionnaire more efficient. The UWIN is essentially a simplified, shorter version of the IPSS and consists of questions about the four most significant lower urinary tract symptoms, with partial scores ranging from 0 to 3, instead of 0 to 5. The final maximum score is 12, instead of 35, the simplified score classifies the intensity of symptoms based on the score into three degrees: mild 0 to 3, moderate 4 to 7, and severe 8 to 12. As with the IPSS, there is an additional simplified question regarding the quality of life (*Crawford et al., 2011*).

The reduction of items in an instrument may cause it to lose sensitivity while potentially gaining specificity, which allows for the detection of more severe cases, even at the expense of failing to recognize some of them. These properties may be considered acceptable when aiming to identify more severe cases for further evaluation. Also, these features are particularly desirable in the primary care setting when limited resources are available, and a considerable proportion of subjects have low health literacy level.

The UWIN has been demonstrated to be a valid and reliable tool for the evaluation of LUTS in men (*Crawford et al., 2011*; *Eid et al., 2014*; *Mallya et al., 2017*; *Ramaraju et al., 2016*), lessening the respondent burden and enhancing the applicability of the IPSS (*Johnson et al., 2008*). The aim of the present study was to perform the cross-cultural adaptation to Brazilian Portuguese and validation of the Urgency, Weak Stream, Incomplete Emptying, and Nocturia (UWIN) for use in Brazil.

## METHODS

The process of cross-cultural adaptation followed the most stringent guidelines on translation and adaptation of health status measures (*McDowell, 2006*; *Beaton et al., 2000*). Followed the recommended steps to achieve conceptual, between-item, semantic, operational, and pretest equivalence to obtain a Brazilian Portuguese version intended for local use. The instrument validity was subsequently tested through the administration of the questionnaire to men living in the metropolitan region of Feira de Santana, Bahia, Brazil, in 2018. To guide the execution of this stage, we followed the quality assessment tool for diagnostic accuracy studies QUADAS-2 (*Whiting et al., 2011*) and the international consensus on health-related patient-reported outcomes, COSMIN (*Mokkink et al., 2010*).

### Stage 1: cross-cultural adaptation

Because the instrument includes clinical components, the initial translation was carried out individually by a committee composed of 3 health professionals to obtain the translated versions T1, T2, and T3. The members of the committee were urologists proficient in the English language and familiar with the topic in question. Following the process of individual translation, a synthesis of the three translations was produced (T4). This version was reassessed by an expert committee, which resolved discrepancies by consensus to create the first Brazilian Portuguese version of the survey (T5), which was, in turn, translated back into the English language (BT1) by a bilingual Brazilian physician. The back-translation (BT1) was again retranslated into Brazilian Portuguese (T6) by the expert committee and evaluated by two medical practitioners. In the absence of alterations, this was considered the adapted version of the survey for pilot testing.

The UWIN in its pretest version (T6) was self-administered to 50 participants so that interpretation issues could be identified. A researcher was responsible for appraising the adequacy of the instructions and the adopted scoring system. The decision about the number of participants included in this stage confirmed to the literature (*Coluci, Alexandre & Milani, 2015*).

Finally, operational equivalence was assessed in the pilot testing stage. In this study, the relevance, adequacy, and format of the instrument, the instructions on how to use it, its modes of administration, scenarios, and categorization were discussed by the expert committee and the research group to reach consensus and improve results. In the absence of modifications, the final version (T6) was termed **UWIN-Br** and utilized for the validation stage.

### Stage 2: validation

The study sample was a convenience sample of 306 consective men older than 40 years included between January and June 2018.

Eligible participants were approached at the clinic reception and invited to participate in the study by an adequately trained researcher. The questionnaire was completed individually in a private location, but the participants were able to ask a researcher for help in case of need. As for illiterate participants and those who had difficulty completing

it on their own, the survey was administered by a researcher in the form of a structured interview. This number is superior to the mean sample size of similar studies.

Both the IPSS and the UWIN-Br, both in Brazilian Portuguese, were utilized for data collection. The total score for each instrument was recorded for each patient to later classify them as having mild, moderate, or severe symptoms. An IPSS score of 0 to 7 indicates mild symptoms, 8 to 19 indicates moderate symptoms, and 20 to 35 indicates severe symptoms. For UWIN-Br, the total score ranges from 0 to 12, with mild, moderate, and severe categories being 0 to 3, 4 to 7 and 8 to 12, respectively.

For the measurement of completion times, a researcher used a stopwatch and recorded the time in minutes individually spent on the task. Uroflowmetry data were obtained from the electronic patient records.

This project was approved by the Research Ethics Committee of the State University of Feira de Santana under the protocol number 64704017.7.0000.0053, position statement 2.052.761 (ANNEX D) and all subjects provided written informed consent.

## Data analysis

Collected data were inputted into two databases to allow for the cross-checking of information and the detection of typing errors. The Statistical Package for the Social Sciences (SPSS) software for Windows, version 22.0, was used for these purposes. GraphPad Prism, version 8.02, was used for data analysis.

Quantitative variables were expressed as medians and interquartile ranges, while qualitative variables were expressed as absolute values, percentages, or proportions (*Vickers & Sjoberg, 2015*).

The total IPSS (0 to 35) and UWIN-Br (0 to 12) scores were determined for each participant, and Spearman correlation coefficients were used to assess criterion validity. As the correlation coefficients measure only the strength of the linear relationship between the two scores, we also used Bland-Altman analysis (*Bland & Altman, 1990*) to determine if the IPSS and UWIN-Br scores were in agreement with each other. The Bland-Altman plot represents the average of the two scores (IPSS + UWIN-Br)/2 versus the difference between them (IPSS–UWIN-Br) for each participant. This method requires the scores of both instruments to be expressed on the same scale. Since the IPSS score ranges from 0 to 35 and the UWIN-Br ranges from 0 to 12, each index was divided by its maximum value, and each score was normalized to values between 0 and 1.

Uroflowmetry was used as a reference standard for the construct validity analysis of UWIN-Br through hypothesis testing. The urinary flow rate was expected to decrease as the score increased, i.e., for patients with more severe symptoms. The maximum urinary flow was determined by uroflowmetry for each of the three categories of symptom severity, then ANOVA was used to compare these data and evaluate the between-group differences and linear trends. We also assessed the diagnostic accuracy of UWIN-Br for the detection of severe cases as defined by the gold standard (IPSS).

Time spent completing the survey (completion time) and need for assistance (for both self-completed questionnaires and interviews) were recorded as a measure of the respondent burden associated with each instrument (UWIN-Br and IPSS)

**Table 1 General and demographic characteristics.**

|  | Median [IQR] |
| --- | --- |
| Age (years) | 59[52–66] |
| Schooling (years) | 11[8–13.7] |
| IPSS score | 6[3–12] |
| UWIN-Br score | 3[1–5] |
| Marital status n (%) |  |
| Not married | 35(11.4%) |
| Married | 247(80.7%) |
| Divorced | 20(6.5%) |
| Widower | 4(1.3%) |

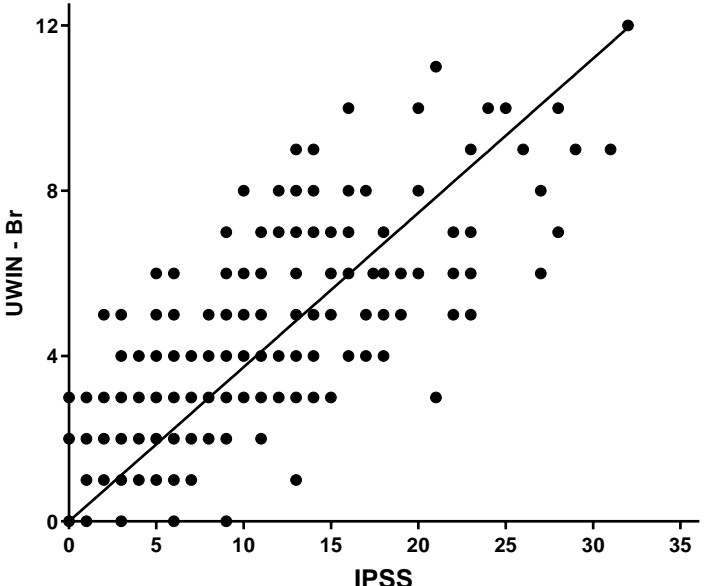

**Figure 1 Spearman correlation between the IPSS and the UWIN-Br.**

(*Andresen, 2000*; *Scientificc Advisor Committee, 2002*). *P*-values <0.05 were considered statistically significant.

## RESULTS

The final study sample comprised 306 men with a median age of 59 [52–87] years. General and demographic data are detailed in Table 1. There was a positive correlation ($r = 0.81$; $p < 0.0001$) between the total IPSS score and the total UWIN-Br score (Fig. 1), which were also positively correlated regarding quality of life ($r = 0.76$; $p < 0.001$). Examination of the

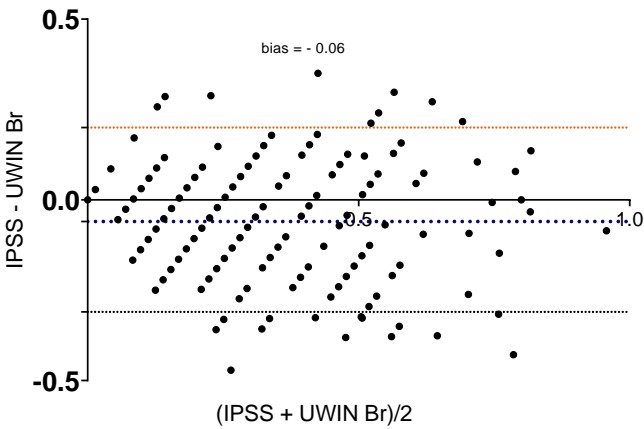

**Figure 2 Bland-Altman plot shows agreement between the IPSS and the UWIN-Br.**

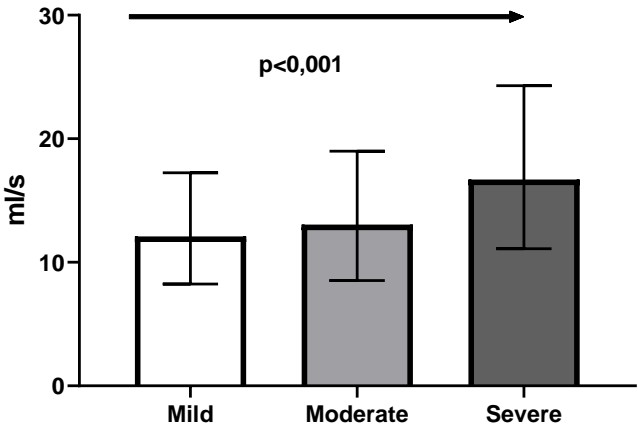

**Figure 3 The relationship between symptom severity according to the UWIN-Br and maximum urinary flow rate (ml/s) determined by uroflowmetry, exhibiting a linear trend (arrow).**

instrument's ceiling and floor effects indicates that 9.4% of patients had lower scores (floor effect) and 1.9% had higher scores (ceiling effect).

Bland-Altman analysis showed good agreement between the two questionnaires and UWIN-Br scores were on the whole slightly higher (Fig. 2), with bias of 0.06 ± 0.13 (bias ± SD). Maximum urinary flow rate values decreased in inverse proportion to the severity of symptoms: 12 [8–16] ml/s in severe cases, 13 [10–19] ml/s in moderate cases and 17 [11–24] ml/s in those with minimal symptoms ($p$ for trend <0.001) (Fig. 3).

UWIN-Br had excellent diagnostic accuracy in detecting the most severe cases, with an area under the ROC curve of 89% (95% CI: [81–97%]; $p < 0.001$).The diagnostic properties are detailed in Table 2. The cut-off value of ≥8 points yielded a sensitivity of 50% and a specificity of 96%, which represent a negative predictive value of 95% and a positive predictive value of 52% in this scenario.

**Table 2 Diagnostic properties of UWIN-Br score.**

| UWIN Score | Sensitivity% | 95% CI | Specificity% | 95% CI |
|---|---|---|---|---|
| 1 | 96,15 | 80,4% to 99,9% | 9,9 | 6,7% to 14,1% |
| 2 | 96,15 | 80,4% to 99,9% | 29,4 | 24,2% to 35,1% |
| 3 | 96,15 | 80,4% to 99,9% | 43,9 | 38,1% to 49,9% |
| 4 | 92,31 | 74,9% to 99,1% | 62,7 | 56,8% to 68,4% |
| 5 | 92,31 | 74,9% to 99,1% | 75,5 | 70,1% to 80,4% |
| 6 | 80,77 | 60,6% to 93,4% | 84,4 | 79,6% to 88,4% |
| 7 | 65,38 | 44,3% to 82,8% | 91,8 | 88,0% to 94,7% |
| 8 | 50 | 29,9% to 70,1% | 95,7 | 92,6% to 97,7% |
| 9 | 42,31 | 23,3% to 63,1% | 98,2 | 95,9% to 99,4% |
| 10 | 26,92 | 11,5% to 47,8% | 98,9 | 96,9% to 99,7% |
| 11 | 15,38 | 4,3% to 34,8% | 99,3 | 97,4% to 99,9% |
| 12 | 11,54 | 2,4% to 30,1% | 99,7 | 98,1% to 99,9% |

Median completion time was 1.2 [1.0–1.5] minute for UWIN-Br and 2.8 [2.2–3.4] minutes for IPSS ($p < 0.001$). Of the total 306 participants, 280 subjects (91.5%) managed to complete the questionnaires with no help, while the other 26 (8.5%) had to be interviewed. The group of participants who needed assistance were significantly older (72 [62–74] versus 58 [51–64] years; $p < 0.001$) and had a lower education level (4 [2–7] versus 11 [8-14] years of education; $p < 0.001$).

## DISCUSSION

Patient Reported Outcome Assessments (PROs) are useful tools for characterizing symptom burden and health-related quality of life (QoL). They are becoming increasingly relevant in the context of clinical and epidemiological research but also in clinical decision-making. Most questionnaires are developed in English (*Vickers & Sjoberg, 2015*), and their proper use in other languages requires a systematic and standardized process of cultural adaptation and validation. This is a necessary demand to assure the appropriateness of the new version of the questionnaire, considering the linguistic and cultural differences (*Crawford et al., 2011*).

The UWIN is a validated instrument for the diagnosis of LUTS (*Eid et al., 2014*), comprising five questions (*Barqawi et al., 2011*; *Crawford et al., 2011*). It has been developed as a simplified questionnaire based on the International Prostate Symptom Score (IPSS). The IPSS is the most used questionnaire for the evaluation of LUTS in men, but subjects with low education level have been shown to have difficulties in completing it accurately (*Johnson et al., 2008*; *Barry et al., 1992*). Following the original publications describing the development (*Barqawi et al., 2011*) and validation (*Crawford et al., 2011*) of the UWIN, other studies used the simplified tool to evaluate its validity and applicability compared to the IPSS (*Eid et al., 2014*; *Mallya et al., 2017*; *Ramaraju et al., 2016*). The UWIN provide results comparable to the IPSS while using a more straightforward format and taking less time to complete and needing lesser assistance even when completed by subjects with lower educational status.

When comparing the UWIN-Br with the IPSS, we found a strong and positive correlation. Also, a high level of agreement was demonstrated in the Bland-Altman plot, indicating that the UWIN-Br has diagnostic properties that are similar to those of the IPSS. Our results are nearly identical to those of the original study and of subsequent validations reported by *Crawford et al. (2011)*, *Eid et al. (2014)* and *Mallya et al. (2017)*, which reinforces the notion that the UWIN is a reliable shorter alternative to the golden standard. There was good agreement between the quality of life measured by the IPSS and the one measured by the UWIN-Br.

The high specificity yielded by the UWIN-Br survey with a cut-off score of ≥8 points (classified as severe) enables this instrument as a useful screening tool, as it neglects those at lower risk and allows for the selective referral of individuals at higher risk to specialized, more complex care.

To our knowledge, this is the first study to validate this instrument in Latin America and the first to adapt it to Brazilian Portuguese. Also moreover, we are the first to compare the UWIN scores with uroflow. This was an essential accomplishment of the present study as it represents the first time the construct validity of the questionnaire was evaluated in comparison to an objective and widely accepted measure of LUTS. Although its use has been recommended by the original author of the instrument (*Crawford et al., 2011*), no studies had performed such evaluation previously.

Uroflowmetry measures the urinary flow rate over time. It is a noninvasive evaluation and represents the most widely used urodynamic test for the assessment of the voiding function. It has been used in most studies evaluating medical and surgical treatments for LUTS-BPH and is used as inclusion criteria for most of these studies. The maximum urinary flow rate (Qmax) is the most widely used variable, despite to have moderate accuracy. Uroflowmetry with a maximum flow rate of <10 ml/s was reported to have lower median sensitivity and specificity of 68% and 70%, respectively (*Malde et al., 2016*).

Most patients answered the questionnaire by themselves without assistance. The comprehension analysis demonstrated that the UWIN-Br is a self-applicable questionnaire, simple and easy to answer.

LUTS are shown to be very common, especially among older patients (*Irwin et al., 2006*; *Van Den Eeden et al., 2012*; *Gravas et al., 2015*). In a USA population-based study, the general prevalence of LUTS was 28%, of which 40% had moderate or severe symptoms (*Glasser et al., 2007*). Another study using a telephone-based survey in Canada, Germany, Italy, Sweden, and the UK found the general prevalence of LUTS to be 62.5% (*Irwin et al., 2006*).

A recently conducted study in five large Brazilian cities found the prevalence of LUTS in men older than 40 years of being 69%. About 30% of participants had severe symptoms, and 40% were very dissatisfied with their condition (*Soler et al., 2017*). In our study, we found the prevalence of moderate and severe symptoms to be 40% in the studied age group. These differences can be explained for two reasons: Our population is older, and our sample was collected in a urological clinic, which adds possible selection bias.

Simplified questionnaires have been recommended in the clinical practice of primary care health professionals as screening instruments, particularly for patients with known

risk factors, to aid in the stratification of a problem and the subsequent investigation into the potential worsening of detrimental health issues such as alcohol consumption and depression (*Zheng et al., n.d*; *Le & Dubertret, 2013*; *Smith et al., 2009*). The adoption of simplified instruments to detect LUTS in primary care settings has also been reported(*Martin et al., 2011*; *Sahai et al., 2014*; *Kajimotu & Bowa, 2018*).

In primary care settings with limited resources and serving patients with a low education level, a common scenario in Brazil, we believe that it is common sense and that health authorities should recommend the following path. First, the simplified instrument should be used to stratify the symptoms, and then the selected subjects recognized to be at higher risk would be referred to specialists according to appropriate, specific guidelines (*Zheng et al., n.d*).

In accord with other studies, the UWIN-Br survey was completed more rapidly and was preferred by a significant proportion of respondents, who found it easier to understand (*Mallya et al., 2017*). Its applicability, simplicity in describing the symptoms and diagnostic properties enable the UWIN-Br as an alternative to the IPSS and may warrant its comprehensive implementation in primary care settings. We believe the actions of this kind would considerably benefit men's health, particularly in the screening of more severe cases. Simplified instruments, like UWIN-Br, reduce respondent burden and increase applicability, optimizing data collection18.

Further research is needed to evaluate how this potential advantage of the UWIN-Br in primary care settings with a low prevalence of severe cases and scarce resources could be reproduced in the longitudinal assessment of symptoms, in the evaluation of new therapies and the detection of LUTS in different age groups and women.

Cross-cultural adaptation and validation of UWIN-Br for use in Brazil seemed relevant to us, as the standardization and implementation of simplified self-report instruments that can be understood by the greatest number of people regardless of education level will in time allow for the adoption of strategies designed to bring men closer to healthcare services in Brazil, mainly in primary care. Such actions are instrumental in expanding and consolidating the Brazilian National Policy of Men's Health Integral Care to lessen health issues among men.

## CONCLUSION

The UWIN-Br has adequate psychometric properties and achieves results similar to those of the IPSS in the evaluation of LUTS and quality of life, but in a simplified format and with better applicability. These features increase the application possibilities of this instrument, especially in the context of men's health actions, enabling the early detection of men at risk of LUTS worsening and their timely referral to specialized care. The UWIN-Br seems appropriate for use in both clinical and research settings

### Funding
The authors received no funding for this work.

### Competing Interests
The authors declare there are no competing interests.

### Author Contributions
- Caroline S. Silva conceived and designed the experiments, performed the experiments, analyzed the data, prepared figures and/or tables, authored or reviewed drafts of the paper, and approved the final draft.
- Katia S. Freitas conceived and designed the experiments, performed the experiments, analyzed the data, authored or reviewed drafts of the paper, and approved the final draft.
- Anna Paloma R. Ribeiro performed the experiments, authored or reviewed drafts of the paper, and approved the final draft.
- Cristiano M. Gomes analyzed the data, prepared figures and/or tables, authored or reviewed drafts of the paper, and approved the final draft.
- Jose Bessa Junior conceived and designed the experiments, analyzed the data, prepared figures and/or tables, authored or reviewed drafts of the paper, and approved the final draft.

### Human Ethics
The following information was supplied relating to ethical approvals (i.e., approving body and any reference numbers):

This project was approved by the Research Ethics Committee of the State University of Feira de Santana (protocol number 64704017.7.0000.0053, position statement 2.052.761).

### Data Availability
The following information was supplied regarding data availability: Data are available as a Supplemental File.

### Supplemental Information
Supplemental information for this article can be found online at http://dx.doi.org/10.7717/peerj.9039#supplemental-information.

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
