# Peer review of "Transcultural adaptation and validation of the questionnaire “Urgency, Weak stream, Incomplete emptying and Nocturia (UWIN)” for the Brazilian Portuguese"

_PeerJ, doi:10.7717/peerj.9039_

## Round 0.1 · original submission · Major Revisions

Dear authors,

After assessing the comments of the reviewers, I think your manuscript has scientific merit to be published in PeerJ, as long as you address the issues highlighted by the reviewers. Please, see their comments below so as to have more information.

Best regards,
Dr Palazón-Bru (academic editor for PeerJ)

Reviewer 1 ·

Basic reporting

1. The English language should be improved. Prosfessional English translator should be consulted.
2.Reference is ok.
3.Figures or tables are good.

Experimental design

1. Research object and question is clear.
2. Ethic issue is ok.
3. The numbers of sample size should be calculated.

Validity of the findings

1. The conclusion is clear.

·

Basic reporting

The authors have justified in detail the need to carry out the work of transcultural adaptation of the UWIN instrument, and it has relevant practical and socio-sanitary implications.
It is a well-written English manuscript with an adequate structure. Bibliographic references used to justify this research are relevant and updated. These are correctly written following the formal recommendations of the journal.

Experimental design

The research question is relevant and well defined.
The article has a rigorous methodology in relation to the translation and validation process. It has been carried out following the COSMIN indications, which ensures alignment with the basic principles of validation studies.
The use of external and objective information such as the urinary flow study constitutes a strong point of this work not previously carried out to assess the criterion-validity and the diagnostic value of the instrument.

Validity of the findings

From my point of view, the results of this research have practical relevance for severe cases of LUTS screening, and as described in the manuscript it can serve as a first-level tool in populations/communities with few resources in Brazil.
Absolutely in accordance with its use as an easy and simple screening tool, prior to other more complex and expensive clinical tests.
The conclusions are clear and respond to the objectives set.

Additional comments

First of all, I would like to congratulate the author on the work. It has a practical application and relevance. The text is clearly written, and it is well understood, despite the specific terminology of the valuation studies.
I would like to make some suggestions that could help the potential reader improve their understanding of the text:
- First, I consider necessary a more detailed description of the instrument and its interpretation as well as the metric properties of other validated versions. For example, it appears in step 2 of the methodology that a score above 8 in UWIN-Br is considered "serious." I believe that I should explain it previously in the introduction when describing the instrument.
- It would be interesting to show the descriptive data of the sample, such as educational level, as you indicate in the discussion that it can be a relevant reason for the work. There are no data on body composition, comorbidities, etc. that are interesting to characterize the sample. I CANNOT find it in table 1 as you indicate. Please, revise.
- Could you report the possible ceiling or floor effects on the different items of the instrument? I have not found any information about it in the manuscript
- You should report the values of the Measurement Error (SEM; MDC…) to know the thresholds of the error and its potential influence on the interpretation of the scores. This is relevant information for readers.
- Line 257. Put reference 18 in hyper index format.
- It would also be interesting to offer the results of a Bland-Altman graph of the UWIN instrument test-retest, in order to know the temporal stability of the scores.

·

Basic reporting

I commend the authors for their extensive data set and analysis. In addition, the manuscript is clearly written in professional, unambiguous language.

Experimental design

The study did achieve its stated aims and objectives

Validity of the findings

The study is well designed

·

Basic reporting

This is adequate

Experimental design

These are adequate

Validity of the findings

This is adequate

Additional comments

The papers is well written and well presented. The message is not new and has been published before. The validation of the UWIN -br questionnaire is well described. It maybe useful for the author to say something about the validation of the IPSS-br which is used in the study. It is noteable that while the Spearmans correlation of 0.804 is given , the next most commonly used measure the sensitivity and specifity of UWIN-br score against IPSS is not given.This would enhence the results. The ROC of 89% has no corresponding results table.The data on the uroflometry appears to have been underutilised in this study and could strengthen further evidence of the accuracy of the UWIN-br. The duration of the IPSS in contrast to the UWIN-br should have p value computed, as well as patient acceptability for both IPSS and the UWIN -br is not given.The study focuses on quantitative data mainly, and says little on the qualitative data which is key in these types of studies.Overall this is a well written paper, with good scientific rigor. these addition would serve to improve it further

---

## Round 0.2 · Minor Revisions

Your work still needs some changes before we can make the final decision.

Reviewer 1 ·

Basic reporting

1. Please consider to change the term ' International Prostatism Symptom Score' in Line 22 to 'International Prostate Symptom Score'
2. Please check the sentence after the full stop in Line 88.[2000). followed the recommended steps..........']. Is it a sentence? if yes, the first letter after full stop should be a block letter.

Experimental design

no comment

Validity of the findings

no comment

·

Basic reporting

The English writting has been reviewed satisfactorily. References are adequate and have been added some relevant news.

Experimental design

Author should describe the method to justify the study sample size.

Validity of the findings

The use of this questionnaire is relevant in the clinical practica in Brazil.
It could be interesting to include a user's guide for this new version, in order to improve its applicability.

Additional comments

I congratulate the authors for their review. They have attended the comments made by this reviewer, and hace included new information that provides relevant data for the research.
However, calculating the sample size and not having data on temporal stability can reduce the power of the study.

---

## Round 0.3 · accepted · Accept

All the reviewers' concerns have been correctly addressed. Therefore, your paper has reached the high standards to be published in PeerJ. Congratulations!